# Novel and Simple Method for Quantification of 2,4,6-Trichlorophenol with Microbial Conversion to 2,4,6-Trichloroanisole

**DOI:** 10.3390/microorganisms11092133

**Published:** 2023-08-23

**Authors:** Saki Goto, Taro Urase, Kaito Nakakura

**Affiliations:** School of Bioscience and Biotechnology, Tokyo University of Technology, Tokyo 192-0982, Japan; gotohsk@stf.teu.ac.jp (S.G.);

**Keywords:** 2,4,6-trichloroanisole (TCA), 2,4,6-trichlorophenol (TCP), odor threshold, *O*-methylation

## Abstract

Contamination with 2,4,6-trichloroanisole (TCA) often causes taste and odor (T&O) problems in drinking water due to its low odor threshold concentration. Microbial *O*-methylation of the precursor 2,4,6-trichlorophenol (TCP) would be the dominant mechanism for TCA formation. Simple and rapid measurement of TCP in the low concentration range is necessary to control the problems induced by TCA. In this study, the combination of microbial conversion and instrumental analysis was proposed as a method of TCP quantification. Fungi and bacteria were isolated from various water samples and examined for their ability to produce TCA from TCP. As a result, a strain exhibiting quantitative TCA production and a high growth rate was obtained and named *Mycolicibacterium* sp. CB14. The conversion rate of TCP to TCA by this strain was found to be high and stable (85.9 ± 5.3%), regardless of the applied TCP concentration, although within the range of 0.1–10 µg/L. The limits of detection and quantification for TCP by this proposed method were determined to be 5.2 ng/L and 17.3 ng/L, respectively. By improving the methods, *Mycolicibacterium* sp. CB14 could be used for the quantification of TCP at very low concentration levels, which is sufficient to manage the T&O problem caused by TCA.

## 1. Introduction

The examination of taste and odor (T&O) is usually the consumer’s first measure of assessing the safety of drinking water, although the concentrations of off-flavor compounds are generally too low to pose any adverse effects on human health [1]. Geosmin (GSM) and 2-methylisoborneol (2-MIB) are well-known compounds that cause T&O problems in drinking water [2]. GSM and 2-MIB, which have low odor thresholds of 10 ng/L and 4 ng/L, respectively [3], are derived primarily from cyanobacteria and other algal species in the sources of drinking water [4]. In addition to these prominent compounds, four other T&O compounds (β-ionone, β-cyclocitral, 2-isobutyl-3-methoxypyrazine, and 2-isopropyl-3-methoxypyrazine), produced by algae, cyanobacteria, and other microorganisms, were detected in the lake waters [5]. Sulfur-containing compounds, such as dimethyl disulfide, nitrogen-containing compounds, such as indoles, and several kinds of aldehydes have also been detected in the raw waters that supply drinking water [6]. Haloanisoles, such as 2,4,6-trichloroanisole (TCA), a famous odorant in wine [7], have recently been recognized as a group of important T&O compounds in drinking water [8,9,10]. In fact, TCA was identified as the likely target for a recent event of T&O problems in the US [11]. Unlike other events experienced at the water treatment facility, the TCA event was probably caused by actinomycetes forming biofilms on the filter media in the treatment facility [11]. In order to mitigate these T&O problems by TCA, it is essential to control its formation in water treatment facilities, including in the distribution pipes, although in the case of the T&O problems by algal-origin compounds, such as GSM and 2-MIB, oxidation and/or adsorption are effective removal processes in the treatment facilities [6,12].

It is known that TCA is produced by microbial *O*-methylation of the precursor 2,4,6-trichlorophenol (TCP) [7,13]. Contamination with TCP possibly takes place because of residual fungicides, herbicides, and insecticides, while TCP is a byproduct of disinfection, which is formed in the chlorination process [14]. Monitoring results in the US, with a quantification limit of 19 ng/L, showed that treated wastewater contained 2,4,6-TCP at an average of 56 ng/L [15], thereby suggesting that the treated wastewater can be a noticeable source of TCA in the water environment when TCP is converted to TCA. Indeed, TCA has recently been recognized as an important T&O compound in biologically treated wastewater [16,17].

Measurement of TCP at a low concentration range is needed to control the concentration of TCA in drinking water because TCA has a very low odor threshold concentration, which is below 1 ng/L [3]. The most common method for measuring the concentration of TCP is to use gas chromatography (GC) with several types of detectors: flame ionization detector (FID), electronic capture detector (ECD), and mass spectrometry (MS) [18], although the sensitivity of the TCP detection (a highly polar molecule) is sometimes unsatisfactory. To improve the chromatogram in the GC analysis, TCP is modified by derivatization using N,O-bis-(trimethylsilyl)trifluoroacetamide (BSTFA) [19]. Several methods for the concentration, separation, and use of liquid chromatography-based instruments have been proposed for the highly sensitive analysis of halophenols [20,21,22]. Fabrication of electrochemical sensors has also been considered for the analysis of TCP [23].

Several fungal species belonging to *Paecilomyces*, *Penicillium*, *Mucor*, and *Trichoderma* are known to perform *O*-methylation of the TCP contained in wine [24]. The examination of eleven fungal isolates taken from water environments showed that the highest TCP to TCA conversion rate (40.5%) was observed using an isolate identified as *Aspergillus versicolor* [25]. In addition to fungal species, bacterial species, such as *Acinetobacter*, *Flavobacterium*, Actinobacteria (*Microbacterium*, *Nocardia*, *Rhodococcus*, and *Streptomyces*), and *Pseudomonas* possibly contribute to the formation of TCA in drinking water processes [26]. Another bacterial species *Sphingomonas ursincola* was also shown to be a TCA producer [27]. The maximum reported conversion rates of TCP to TCA were 60% [26]. Excluding the isolates of the pure culture, the reported conversion rates of halophenols to haloanisoles, by a mixed culture system, were reported to be very low [10], probably due to a high variety of degradation pathways by halophenols.

In this study, we succeeded to obtain a bacterial strain that can convert TCP, almost completely, to TCA, even when using a very low concentration range. It was shown that the *Mycolicibacterium* sp. CB14 strain could be used as a simple and novel method for the measurement of TCP, with a quantitative conversion to TCA, which does not require the derivatization step in the GC/MS analysis.

## 2. Materials and Methods

### 2.1. Isolation of Microorganisms

Treated wastewater samples were taken from wastewater treatment plants in 2021, both after chlorination (T1 and T2) and before chlorination (T3). T1 and T3 treated kitchen, toilet, and laboratory wastewater were collected from a University. T2 was a municipal wastewater treatment plant in Tokyo, Japan. River water samples were gathered in 2021 at the Sagami River (R1) in Kanagawa prefecture and at the Onda River (R2) in the suburbs of Tokyo. The samples were also collected from hot water baths in a university dormitory (B1, B2), from a hot water bath in a sports gym (B3), from cold water baths in the same sports gym (B4), and from a swimming pool (B5) in Japan from 2021 to 2022. The water samples were collected in sterilized glass bottles in the morning and immediately brought back to the laboratory for analysis in the afternoon on the same day as the sampling. A small volume (0.1 mL) of the samples was placed and incubated for 7 days at 25 °C on potato dextrose agar plates (Solabia BIOKAR Diagnostics, Pantin, France), supplemented with chloramphenicol at a concentration of 0.5 mg/L, to obtain fungi isolates and to suppress the growth of bacteria. A total of 32 fungi isolates were obtained (Table 1). R2A agar plates (R2A agar DAIGO, Shiotani M.S. Co., Ltd., Amagasaki, Japan) were also used to obtain bacterial isolates following incubation for 7 days at 25 °C; this low nutrient agar is suitable to obtain a high number of bacterial counts for environmental samples [28]. A total of 35 isolates were obtained for bacteria (Table 2).

### 2.2. Identification of the Isolates

#### 2.2.1. Fungal Isolates

The phylogenetic characteristics of the obtained fungal isolates were examined by sequencing the ITS (internal transcribed spacer) region between the 18S and 26S rRNA genes. Before sequencing, DNA was extracted using a DNA extraction kit (Kanto Chemical Co., Ltd., Tokyo, Japan). The extracted ITS rRNA gene was amplified by primers ITS-1F (5′-GTAACAAGGTYTCCGT-3′) and ITS-1R (5′-CGTTCTTCATCGATG-3′). The amplified DNA was purified using the MonoFas DNA clean-up kit (GL Sciences Inc., Tokyo, Japan) and sent to Macrogen Japan Corp. (Tokyo, Japan) for Sanger sequencing, using sequencing primer ITS-1 (5′-TCCGTAGGTGAACCTGCGG-3′). The closest species or genera were determined by comparing the stain types with the BLASTN program (https://blast.ncbi.nlm.nih.gov/ accessed on 21 March 2023), provided by the National Center for Biotechnology Information (NCBI), National Institute of Health, US.

#### 2.2.2. Bacterial Isolates

The phylogenetic characteristics of the obtained bacterial isolates were examined by sequencing the full-length 16S rRNA gene. Before sequencing, DNA was extracted using a DNA extraction kit (Kanto Chemical Co., Ltd., Tokyo, Japan). The extracted 16S rRNA gene was amplified using primers 27F (5′-AGAGTTTGATCMTGGCTCAG-3′) and 1492R (5′ -GGYTACCTTGTTACGACTT-3′). The amplified DNA was purified using the MonoFas DNA clean-up kit (GL Sciences Inc., Tokyo, Japan) and sent to Macrogen Japan Corp. (Tokyo, Japan) for Sanger sequencing, using sequencing primers 518F (5′-CCAGCAGCCGCGGTAATACG-3′) and 800R (5′-TACCAGGGTATCTAATCC-3′). The two obtained sequences were connected to provide a full-length 16S rRNA sequence. The closest species or genera were determined by comparing the sequences within type strains by the BLASTN program (https://blast.ncbi.nlm.nih.gov/ accessed on 21 March 2023), provided by the National Center for Biotechnology Information (NCBI), National Institute of Health, US.

### 2.3. Screening for Producers of TCA from TCP

A bacterial isolate was inoculated in an autoclaved vial bottle filled with 10 mL R2A agar (R2A agar DAIGO, Shiotani M.S. Co., Ltd., Amagasaki, Japan) supplemented with TCP at a concentration of 100 μg/L. After incubation at 25 °C for 5–10 days in the sealed vial bottle, the produced TCA and residual TCP were analyzed by headspace–solid phase micro-extraction (HS-SPME) gas chromatography–mass spectrometry (GC/MS) method [1], with several modifications. In brief, a solid-phase microextraction (SPME), fiber coated with polydimethylsiloxane (PDMS, 100 μm, Supelco), was used before its introduction to GC–MS (GC-2010/parvum 2, Shimadzu, Kyoto, Japan), equipped with a separation column InertCap 5MS/Sil (0.25 mm, 30 m, 0.25 μm (GL Sciences Inc., Tokyo, Japan). The GC injection port in the splitless injection mode was maintained at 260 °C. The gas flow was controlled by constant linear velocity at 42 cm/sec. The column oven temperature was programmed at 40 °C (3 min)–10 °C/min–80 °C–15 °C/min–250 °C (3 min). Both detector and interface temperatures were maintained at 250 °C. The TCA peak was integrated with the target ion 210 *m*/*z* and the reference ion 212 *m*/*z*; the TCP peak was integrated with the target ion 196 *m*/*z* and the reference ion 198 *m*/*z*. The detector voltage was set at +0.3 kV, relative to the automated tuning value. For the fungal isolates, the same procedures were applied as for the bacterial isolates, except that potato dextrose agar (Solabia BIOKAR Diagnostics, France) supplemented with TCP was used at a concentration of 100 μg/L instead of the R2A agar. In addition to the isolates obtained in this study, *Trichoderma longibrachiatum* NBRC 4847 was examined as a known TCA producer from TCP, while *Saccharomyces cerevisiae* INV SC1 and two *Escherichia coli* strains (NBRC 13168 and NBRC 13965) were examined as negative controls in the TCA production analysis.

### 2.4. Quantitative Analysis of the TCP to TCA Conversion by Mycolicibacterium sp. CB14

Liquid R2A broth consisting of polypepton, 0.5 g/L, yeast extract 0.5 g/L, casamino acid 0.5 g/L, glucose 0.5 g/L, soluble starch 0.5 g/L, dipotassium phosphate 0.3 g/L, magnesium sulfate heptahydrate 0.05 g/L, and sodium pyruvate 0.3 g/L was prepared. *Mycolicibacterium* sp. CB14, taken from B4, was selected to quantitatively produce TCA from TCP (0524-S14, accession number: OQ651233) with a high growth rate. After preincubating *Mycolicibacterium* sp. CB14 in the liquid R2A broth for a day, 100 μL of the preincubated strain was added to 10 mL new liquid R2A broth supplemented with TCP at concentrations of 0, 0.1 μg/L, 1 μg/L, 10 μg/L, and 100 μg/L, and then, incubated for another day at 30 °C. Before determining the concentrations of the produced TCA and residual TCP, sodium chloride 2.4 g and *p*-Iodoanisole (an internal standard; concentration of 0.5 μg/L) were added to ensure an accurate measurement by headspace–solid phase micro-extraction (HS–SPME) gas chromatograph–mass spectrometry (GC/MS) method. The SPME fiber, the instrument, and the operational method program for the GC/MS analysis were the same as was previously used in the screening test.

## 3. Results

### 3.1. Isolation of TCA Producer

#### 3.1.1. Fungal Isolates

Figure 1 shows the results of the GC/MS analysis on the TCA production and residual TCP in the screening test by the fungal isolates. The detailed results, including the appearances of the colonies, registered accession numbers, and identified genera, are shown in Appendix A. The results of the TCA and TCP concentrations are shown as relative peak areas compared to those obtained for the reference vial bottles containing the potato dextrose agar supplemented with TCP. *Trichoderma longibrachiatum* NBRC 4847, a known TCA producer [29,30], showed a high TCA production and a lower concentration of residual TCP, while *Saccharomyces cerevisiae* INV SC1 presented a negative result. Among the isolates in this study, all the *Trichoderma* isolates showed high TCA productions, followed by two isolates out of the three *Penicillium* spp., one isolate out of the two *Talaromyces* spp., and one isolate out of the three *Trametes* spp. These were consistent with a previous observation [25], where the genera (*Trichoderma*, *Talaromyces*, *Penicillium*, and *Trametes*) that produced high levels of TCA had filamentous morphologies under microscopic observation. In spite of the high consumptions of TCP by *Aspergillus* spp. and *Trametes* spp., the concentrations of TCA after incubation were very low, possibly because TCP was converted to other compounds, except for TCA, by these fungal isolates. Three *Cladosporium* spp. isolates produced negligible TCA despite relatively high consumptions of TCP. *Geotrichum* sp., *Rhodotorula* spp., and all isolates belonging to Tremellomycetes (*Naganishia* spp., *Apiotrichum* spp., *Cutaneotrichosporon* spp., and *Trichosporon* spp.) were incapable of consuming TCP, which resulted in negative TCA productions.

#### 3.1.2. Bacterial Isolates

Figure 2 shows the results of the GC/MS analysis of the TCA production and residual TCP in the screening test of the bacterial isolates. The detailed results, including the appearances of the colonies, registered accession numbers, and identified genera, are shown in Appendix A. The results on the TCA and TCP concentrations are shown as the relative peak area to those obtained for reference vial bottles containing R2A agar and supplemented with TCP. Two *E. coli* strains (NBRC 13168 and NBRC 13965) showed negative results. Among the 35 isolates in this study, the isolates with high TCA productions were concentrated on the Actinomycete members, which partly reflected the selectivity of the R2A agar and the bacterial population in the samples; moreover, all *Mycolicibacterium* and *Mycobacterium* isolates had high TCA productions. Notably, high TCA productions by actinomycetes is consistent with previous literature [11,26]. On the other hand, most Proteobacteria members, except for one isolate out of five *Sphingomonas* spp., produced negligible TCA. The exception (*Sphingomonas*) is consistent with previous literature, whereby the order Sphingomonadales is a known oxidative degrader of TCP [31] and a known TCA producer [27]. The contribution of Sphingomonadales to the microbial conversion of TCP to TCA would be of importance because of a certain dominance of this species in drinking water distribution systems [32]. Although one isolate out of three *Acinetobacter* spp. showed a small amount of TCA production, and a considerable part of the added TCP remained in the agar. Other isolates, such as *Micrococcus* sp., *Methylobacterium* sp., *Rheinheimera* sp., *Yersinia* sp., *Pseudomonas* spp., and all isolates belonging to Bacilli (*Bacillus* sp., *Exiguobacterium* sp., and *Staphylococcus* sp.), Flavobacteriia (*Flavobacterium* spp. and *Chryseobacterium* sp.), and Betaproteobacteria (*Pelomonas* sp. and *Mitsuaria* sp.) were not capable of consuming TCP, meaning that they were negative for TCA production. The reason for the observed relative TCP peak areas, which were higher than 100%, was the decrease in pH that affected the extraction efficiency in the determination of TCP by the SPME method [33]. Unlike in the case of TCP (acidic molecule), the change in pH would only have a negligible effect on the judgment of TCA production in the screening test.

### 3.2. Production of TCA by Mycolicibacterium sp. CB14

The quantity of TCA that was converted from TCP by *Mycolicibacterium* sp. CB14 was analyzed in this section. The production of the TCA was quantified based on a calibration curve for TCA (*R*^2^ of 0.9986; Appendix A). The microbial conversion of TCP to TCA produced a straight calibration line when the formed TCA was plotted against the feed TCP using an investigated concentration range between 0 and 10 μg/L, with a high correlation (*R*^2^ of 0.9954; Appendix A). *Mycolicibacterium* sp. CB14 quantitatively produced TCA depending on the TCP concentration that was added to the liquid medium (*R*^2^ > 0.99) (Appendix A), although a slight production of TCA was observed in the vial bottle filled with R2A medium without TCP. The slight production of TCA without the addition of TCP was probably caused by the presence of small amounts of TCA precursors contained in the liquid medium and in the intercellular materials of *Mycolicibacterium* sp. CB14. The conversion rates of TCA from TCP by *Mycolicibacterium* sp. CB14 were determined by subtracting the small amount of detected TCA without the addition of TCP (Figure 3). The conversion rates in the medium containing 0.1, 1, and 10 µg/L TCP were 88.1%, 84.3%, and 85.2%, respectively, with a high and stable mean value of 85.9 ± 5.3%. The number of *Mycolicibacterium* sp. CB14 colonies were calculated based on the correlation between colony numbers and the turbidity of the culture medium (Appendix A) and did not affect the conversion rates. The limits of detection (LOD) and quantification (LOQ) were determined to be 5.2 ng/L and 17.3 ng/L, respectively, based on 3 times the SD and 10 times the SD of the peak areas in the GC/MS analysis, where SD represents the standard deviation in the repeated measurements of the blank samples without TCP addition.

## 4. Discussion

The results of the screening tests for TCA-producing fungi and bacteria showed that isolates identified as *Trichoderma* spp., *Penicillium* spp., *Talaromyces* spp., *Trametes* spp., *Mycolicibacterium* spp., *Mycobacterium* sp., and *Sphingomonas* sp. had relatively high TCA productions. In previous studies, the conversion rates of TCA from TCP by *Trichoderma* spp., *Penicillium* spp., and *Talaromyces* spp. isolated from wine corks and drinking water treatment plants were reported in the range of 2.7–37.6% [24,25,34], while bacterial species *Mycobacterium* spp. and *Sphingomonas* spp. have been suggested to be capable of producing TCA [26,35]. On the other hand, a study showed that an isolate of *Trametes versicolor* taken from a drinking water treatment plant, where a high occurrence of TCA was observed, did not show methylation to TCA [25]. *Aspergillus* spp. and *Cladosporium* spp. produced very little TCA in this study, although some studies showed TCA conversion rates of 14.3–40.5% [25,34]. Summarizing these studies, the TCA production capacity differs depending on the species, even within the same genus. Slight genomic differences among strains in the same genus and differences in the inducibilities of *O*-methyltransferase may have affected TCA production. It was also shown that some isolates in this study consumed TCP without TCA accumulation. A possibility that could be considered is that the produced TCA was degraded to other compounds. Indeed, *T. versicolor*, *Pseudomonas putida* INBP1, and *Acinetobacter radioresistens* INBS1 were reported to biodegrade TCA [36,37].

The *Mycolicibacterium* sp. CB14 bacterial strain, isolated in this study from the cold-water baths (B4), showed a high TCA production and grew relatively quickly. The full 16S rRNA gene sequence in this isolate showed 99.9% homology to *Mycolicibacterium phocaicum*. It was reported that *O*-methylation was expressed constitutively without any inducers in several *Rhodococcus* and *Mycobacterium* strains [35], while *O*-methyltransferase purified from *T. longibrachiatum* was induced by several chlorophenols [29]. It was expected that *Mycolicibacterium*, which is a close genus to *Mycobacterium*, with constitutive *O*-methyltransferase could convert TCP to TCA quantitatively, even at a low concentration range. The high and stable conversion rate (88.1%), even in the medium containing 0.1 µg/L TCP, by *Mycolicibacterium* sp. CB14 suggested that the isolate can convert TCP to TCA, even at a very low concentration range of TCA at 0.03–10 ng/L, which are odor threshold levels [3].

In the event of T&O problems, low sensitivities for phenolic compounds were barriers to the swift identification of the compounds responsible for these events [11]. Several attempts have been made to enhance the sensitivity of the analysis of TCP by optimizing extraction techniques, modifying mobile-phase composition, derivatizing TCP, and improving the electrodes [18,22,23,38]. The quantification method used in this study achieved an equivalent LOQ, as has been reported in previous literature (19–10,000 ng/L). To prevent T&O problems in the supply of drinking water, a lower quantification limit (below 1 ng/L) is desirable in the analysis of TCP, which could possibly be converted to the odorant TCA through the treatment process and pipelines if the conditions are appropriate for microorganisms. One of the usual practices to increase the sensitivity of GC analysis is the derivatization of TCP to form a less polar compound using, for example, BSTFA [19].

Instead of chemical derivatization by BSTFA, this study proposes the microbial conversion of TCP to TCA before analysis by GC. By only increasing the sample volume, even in the routine GC/MS analysis, a LOQ of 0.1 ng/L can easily be achieved for TCA [4]. Hence, this study proposes the conversion of TCP to TCA by bacteria before analysis by GC/MS, to assess the production of low concentrations of the odorant. By using *Mycolicibacterium* sp. CB14, which were isolated in this study, TCP could be measured by routine GC/MS analysis at very low concentrations. However, a limitation of the proposed method is the requirement of keeping the bacterial culture for bioconversion in laboratories. Another limitation is the requirement for the removal of bacteria from the samples prior to the TCP analysis to avoid non-intentional bioconversion. To overcome these drawbacks, the use of enzymatic reactions in the substitution of the bacterial reaction could be suggested. A shorter reaction time without removing the bacteria from the samples could also be expected by introducing the enzymatic reaction. Studies on the enzymatic conversion of TCP to TCA will be the next step in this area.

## 5. Conclusions

In this study, fungi and bacteria that are capable of producing TCA were isolated from water samples. Among the 32 fungal and 35 bacterial isolates in this study, 11 and 15 isolates showed an ability to produce TCA, respectively. The fungal isolates in the positive production of TCA were identified as *Trichoderma* spp., *Penicillium* spp., *Talaromyces* sp., and *Trametes* sp., based on the ITS-rRNA sequences, while the bacterial isolates that positively produced TCA were identified as *Mycolicibacterium* spp., *Mycobacterium* sp., and *Sphingomonas* sp., based on the 16S rRNA sequences. This study proposed a method to quantify the precursor TCP by using the microbial conversion of TCP to TCA and improving the chromatogram in GC. Among various isolates in this study, *Mycolicibacterium* sp. CB14 was selected based on its high growth rate and quantitative conversion rate from TCP to TCA (85.9%), regardless of the applied TCP concentration (0.1–10 µg/L). The proposed quantification method achieved a LOD of 5.2 ng/L and LOQ of 17.3 ng/L. By improving the method, *Mycolicibacterium* sp. CB14 could be used to quantify TCP at very low concentration levels, which is sufficient for the management of T&O problems caused by TCA.

## Figures and Tables

**Figure 1 microorganisms-11-02133-f001:**
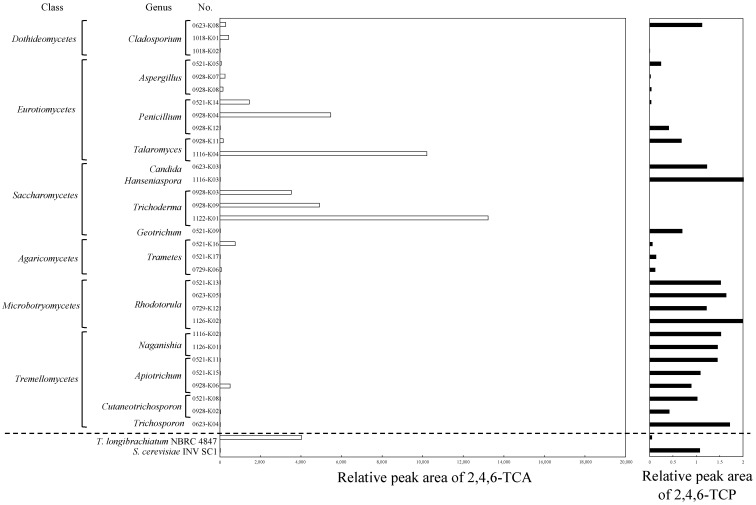
The results of the GC/MS analysis on TCA production and residual TCP in the screening test by the fungal isolates.

**Figure 2 microorganisms-11-02133-f002:**
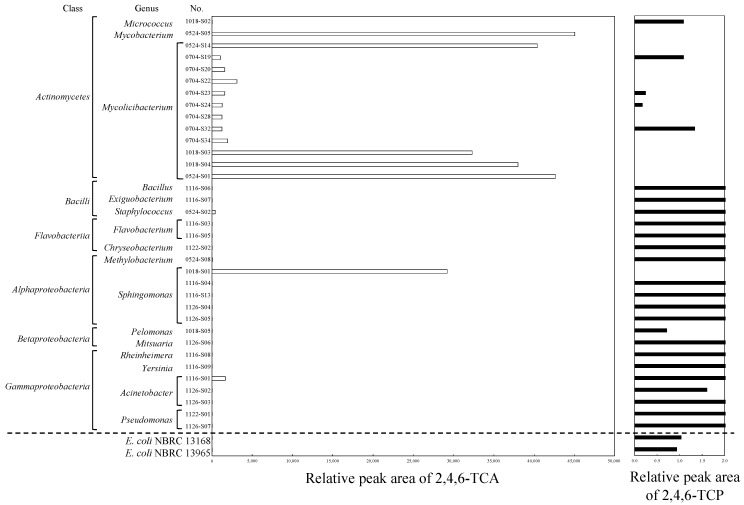
The results of the GC/MS analysis on the TCA production and residual TCP in the screening test by the bacterial isolates.

**Figure 3 microorganisms-11-02133-f003:**
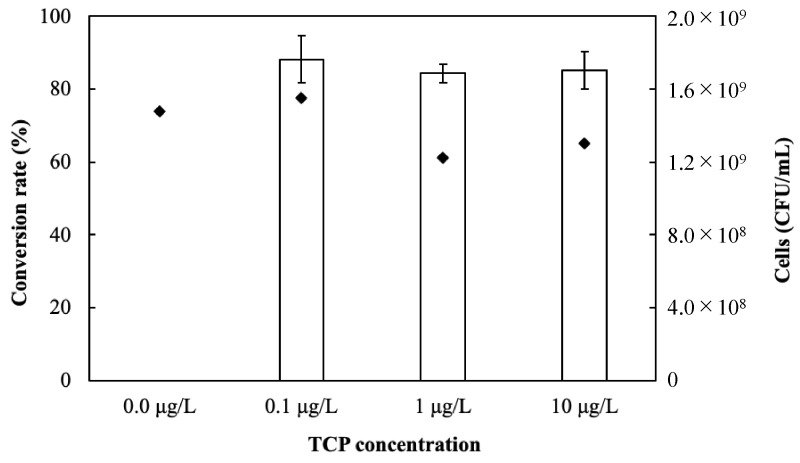
The conversion rates from TCP to TCA and the number of *Mycolicibacterium* sp. CB14 colonies in each TCP concentration. White bar: conversion rate of TCP to TCA; diamond: number of colonies.

**Table 1 microorganisms-11-02133-t001:** Origins of the samples and the numbers of fungal isolates obtained from the samples.

Sampling Dates	Sampling Locations	Number of Isolates
21 May 2021	T1	9
20 June 2021	T2	4
28 July 2021	T2	2
28 September 2021	T3	9
18 October 2021	B1	2
16 November 2021	R1	3
20 November 2021	R2	1
26 November 2021	B2	2

**Table 2 microorganisms-11-02133-t002:** Origins of the samples and the numbers of bacterial isolates obtained from the samples.

Sampling Dates	Sampling Locations	Number of Isolates
18 October 2021	B1	5
16 November 2021	R1	9
20 November 2021	R2	2
26 November 2021	B2	6
16 May 2022	B3	3
16 May 2022	B4	2
26 Jun 2022	B5	8

## Data Availability

Not applicable.

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
