# Peer review of "Novel and Simple Method for Quantification of 2,4,6-Trichlorophenol with Microbial Conversion to 2,4,6-Trichloroanisole"

_microorganisms, 2023, doi:10.3390/microorganisms11092133_

Round 1

Reviewer 1 Report

The article “Novel and simple method for quantification of 2,4,6-trichlorophenol with microbial conversion to 2,4,6-trichloroanisole” submitted to Microorganisms by Goto et al. screened the fungi and bacteria capable of converting TCP to TCA and demonstrated a potential way to quantify TCA without the involvement of derivatization. The experiment was well-designed, and the proposed method is innovative. However, I have a few questions on this manuscript, especially regarding the applicability of this method.

1. L 84-86, why did authors add chloramphenicol in the Potato-dextrose agar plates, but not to R2A agar plates?

2. Since the authors discussed the essential parameters for TCA and TCP detection using GC-MS, I recommended including other parameters of this method (e.g., inlet temperature, column flow rate, collision gas flow, ion source temperature) in supplementary materials.

3.  L 165-166, this sentence has a minor grammatical error.

4.  L 196, could the pH affect the extraction efficiency of TCA as well? Would it be appropriate to slightly adjust the pH before performing the SPME method to mitigate this potential influence?

5. Section 3.2, why did the authors not choose Mycobacterium to be the candidate for this method, as it showed highest TCA intensity?

6. The proposed method for detecting TCA through microbial conversion is intriguing. Did the authors assess its accuracy of this method in real water samples, considering potential interference from other factors such as other bacteria, as mentioned in L 63?

Minor editing of English language required.

Author Response

The article “Novel and simple method for quantification of 2,4,6-trichlorophenol with microbial conversion to 2,4,6-trichloroanisole” submitted to Microorganisms by Goto et al. screened the fungi and bacteria capable of converting TCP to TCA and demonstrated a potential way to quantify TCA without the involvement of derivatization. The experiment was well-designed, and the proposed method is innovative. However, I have a few questions on this manuscript, especially regarding the applicability of this method.

Reply: Thank you for your review. We have revised the manuscript to reply to your valuable comments and suggestions.

  1. L 84-86, why did authors add chloramphenicol in the Potato-dextrose agar plates, but not to R2A agar plates?

Reply: The objective of the addition of chloramphenicol is to suppress the growth of bacteria. Chloramphenicol was not added to R2A, because R2A agar was used to obtain bacterial isolates. To improve the manuscript, the words “to suppress the growth of bacteria” were added to the manuscript in the section “2.1 Isolation of microorganisms”

  1. Since the authors discussed the essential parameters for TCA and TCP detection using GC-MS, I recommended including other parameters of this method (e.g., inlet temperature, column flow rate, collision gas flow, ion source temperature) in supplementary materials.

Reply: The details for GC/MS analysis was added in the section of “2.3 Screening for producers of TCA from TCP”, by writing “The injection port of GC in the splitless injection mode was maintained at 260oC. The gas flow was controlled with constant linear velocity at 42 cm/sec. The column oven temperature was programmed at 40 oC (3 min) – 10 oC/min – 80 oC – 15 oC/min – 250 oC (3 min). Both detector and interface temperatures were maintained at 250oC. The peak for TCA was integrated with the target ion 210 m/z and the reference ion 212 m/z; That for TCP was integrated with the target ion 196 m/z and the reference ion 198 m/z. The detector voltage was set at + 0.3 kV relative to the automated tuning value.”

  1. L 165-166, this sentence has a minor grammatical error.

Reply: The error in the section “3.1.1. Fungal isolates” has been corrected, by writing “In spite of high consumptions of TCP by Aspergillus spp. and Trametes spp., the concentrations of TCA after incubation were very low, possibly because TCP was converted to other compounds except for TCA by these fungal isolates.”

  1. L 196, could the pH affect the extraction efficiency of TCA as well? Would it be appropriate to slightly adjust the pH before performing the SPME method to mitigate this potential influence?

Reply: In the section of “3.1.2. Bacterial isolates”, the authors wrote in the revised manuscript “Unlike the case of TCP (acidic molecule), the change in pH would have only a negligible effect on the judgment of TCA production in the screening test.”

  1. Section 3.2, why did the authors not choose Mycobacterium to be the candidate for this method, as it showed highest TCA intensity?

Reply: Not only the TCA production, but the growth rate was taken into account for the selection of the isolates. For a better understanding, the authors added “and to have a high growth rate.” in the section of “2.4. Quantitative analysis on the conversion from TCP to TCA by Mycolicibacterium sp. CB14”. In the abstract and in the conclusion, the same phrases were added to keep the consistency.

  1. The proposed method for detecting TCA through microbial conversion is intriguing. Did the authors assess its accuracy of this method in real water samples, considering potential interference from other factors such as other bacteria, as mentioned in L 63?

Reply: The authors did not complete the studies on the application of this methods to analyze environmental samples. To clarify the direction of this research, the authors added “One of the limitations of the proposed method is the requirement of keeping the bacterial culture for the bioconversion in laboratories. Another limitation is the requirement of the removal of bacteria from the samples for the analysis of TCP to avoid non-intentional bioconversion. To overcome these drawbacks, the use of enzymatic reaction in substitution of bacterial reaction could be suggested. A shorter reaction time without removing bacteria from the samples will be expected by introducing the enzymatic reaction. Studies on the enzymatic conversion from TCP to TCA will be the next step of this study.” at the end section of the chapter of discussion.

Reviewer 2 Report

1.     Although TCP is a precursor of TCA, there are certain conditions and conversion efficiency for TCP to TCA. Therefore, detecting TCP does not seem to be directly related to TCA. Therefore, the significance of this study is not clear enough. That is to say, what’s the meaning of “Mycolicibacterium sp. CB14 could be used 19 for the quantification of TCP at a very low concentration level below 1 ng/L (L19-20).”

2.     Although the author has found a bacterial strain that can convert TCP into TCA completely, is the bacterial strain present in the environment everywhere? If there is no Mycolicibacterium sp. (CB14) strain in the water supply network, then this method is ineffective, and therefore, the scope and conditions of application of this method are not clearly stated. That is to say, the innovation and applicability of the methods found in this study need to be further described.

3.     L266 “……through the treatment process and pipelines if the conditions are appropriate for 266 microorganisms.” The relevant conversion effect on the pipeline was not seen in the article, and this sentence is not detailed in this article. Please improve the discussion.

4.     L66 “we succeeded to obtain a bacterial strain which can convert nearly 100% of TCP to TCA even at a very low concentration range.”. However, L256 “The high and stable conversion rate (88.1%) even in the medium……”. L215-216 “The conversion rates in the medium containing 0.1, 1, and 10 μg/L TCP were 88.1%, 84.3%, and 85.2%, respectively,”88.1%, 84.3%, and 85.2% are not equal to 100%. The results and discussion should be more rigorous in this study.

5.     The abstract, discussion, and conclusion are not consistent and need to be further refined and improved.

6.     L224: The font size in Figure 3 is too large, and the font sizes in other figures also need to be adjusted uniformly.

7.     There is a reference 10 in L259, the reference was published in 2013, so the novelty of this study requires in-depth discussion.

8.     Several typos in grammar have been identified. Please double-check the script for language and typos. eg. L60 “contributes” should be “contribute”.

9.     In the introduction part, more than 90% literature is before 2020. So very difficult to judge the novelty of the current work, which is not based on current-state-of-the-art. Must supply some novel and Up-to-date literature.

Please double-check the script for language.

Author Response

Reply: Thank you for your review. We have revised the manuscript to reply to your valuable comments and suggestions.

  1. Although TCP is a precursor of TCA, there are certain conditions and conversion efficiency for TCP to TCA. Therefore, detecting TCP does not seem to be directly related to TCA. Therefore, the significance of this study is not clear enough. That is to say, what’s the meaning of “Mycolicibacterium sp. CB14 could be used for the quantification of TCP at a very low concentration level below 1 ng/L (L19-20).”

Reply: The authors will reply to this comment combining with the reviewer’s third questions.

  1. Although the author has found a bacterial strain that can convert TCP into TCA completely, is the bacterial strain present in the environment everywhere? If there is no Mycolicibacterium sp. (CB14) strain in the water supply network, then this method is ineffective, and therefore, the scope and conditions of application of this method are not clearly stated. That is to say, the innovation and applicability of the methods found in this study need to be further described.

Reply: The authors will reply to this comment combining with the reviewer’s third questions.

  1. L266 “……through the treatment process and pipelines if the conditions are appropriate for microorganisms.” The relevant conversion effect on the pipeline was not seen in the article, and this sentence is not detailed in this article. Please improve the discussion.

Reply: The authors are afraid that the reviewer may have partly misunderstood the method we propose in this study. In this manuscript, we propose to use Mycolicibacterium sp. (CB14) strain on the occasion of the analysis of TCP. We do not propose to use it in the environment. We propose to convert TCP to TCA in laboratories providing an ideal environment for the strain to biologically convert TCP to TCA just before the GC/MS analysis. The authors agree with the reviewer that there are several limitations of the proposed methods. To solve the reviewer’s negative impression to our proposed methods, the authors added “One of the limitations of the proposed method is the requirement of keeping the bacterial culture for the bioconversion in laboratories. Another limitation is the requirement of the removal of bacteria from the samples for the analysis of TCP to avoid non-intentional bioconversion. To overcome these drawbacks, the use of enzymatic reaction in substitution of bacterial reaction could be suggested. A shorter reaction time without removing bacteria from the samples will be expected by introducing the enzymatic reaction. Studies on the enzymatic conversion from TCP to TCA will be the next step of this study.” at the end section in the chapter of discussion.

In addition, by adding “In the event of T&O problem, low sensitivities for phenolic compounds were barriers for the swift identification of the compounds responsible for the event [11].” in the chapter of discussion, the authors expect that the readers understand the occasion when the proposed method can be used.

  1. L66 “we succeeded to obtain a bacterial strain which can convert nearly 100% of TCP to TCA even at a very low concentration range.”. However, L256 “The high and stable conversion rate (88.1%) even in the medium……”. L215-216 “The conversion rates in the medium containing 0.1, 1, and 10 μg/L TCP were 88.1%, 84.3%, and 85.2%, respectively,”88.1%, 84.3%, and 85.2% are not equal to 100%. The results and discussion should be more rigorous in this study.

Reply: The authors deleted “nearly 100%” from the manuscript to comply with the reviewer’s comment.

  1. The abstract, discussion, and conclusion are not consistent and need to be further refined and improved.

Reply: The authors revised the conclusion and the abstract to be more consistent as to (1) the reason for the selection of the strain, (2) LOD and LOQ by the proposed method, and (3) the level of the concentration applied for the proposed method. In the revised manuscript, the conclusion, the discussion and the abstract are revised to keep the consistency.

  1. L224: The font size in Figure 3 is too large, and the font sizes in other figures also need to be adjusted uniformly.

Reply: The authors revised the figure to be smaller. However, the font size in Figure 3, even after the revision, was not the same as is used in Figures 1 and 2, because Figures 1 and 2 contain all results for the screening test and it was difficult to enlarge the font size of these figures. The authors expect the reviewer’s understanding.

  1. There is a reference 10 in L259, the reference was published in 2013, so the novelty of this study requires in-depth discussion.

Reply: In the discussion, the authors changed the reference [10] to a literature [3] published in 2021 as to the odor thresholds, and to another literature [4] published in 2021 as to the detection limits.

  1. Several typos in grammar have been identified. Please double-check the script for language and typos. eg. L60 “contributes” should be “contribute”.

Reply: The authors checked the manuscript carefully again to improve the language.

  1. In the introduction part, more than 90% literature is before 2020. So very difficult to judge the novelty of the current work, which is not based on current-state-of-the-art. Must supply some novel and Up-to-date literature.

Reply: The authors added five new references [3,4,6,11,12] published in 2021 – 2023 to comply with the comments by the reviewer. By these changes, the manuscript has been improved in terms of novelty.

Reviewer 3 Report

Reviewer’s comments:

The manuscript contains the description of the results of original Authors’ approach. To determine the concentration of 2,4,6-trichlorophenol at the concentration level 5.2/17.3 ng/L (LOD/LOQ) in drinking water, its conversion into 2,4,6-trichloroanisol is recommended. However, this transformation is provided not by chemical derivatization, but by microbial conversion. Several promising strains to solve this problem have been identified, e.g., Mycolicibacterium sp. CB-14.

Reading the text did not lead to the revealing of any problems, neither actual, nor stylistic. Let me congratulate the Authors with so professionally presented results.

Author Response

Reply: Thank you for your review on our manuscript. The authors understand that your comments do not suggests revisions on our manuscript.

Round 2

Reviewer 2 Report

The authors have carefully revised the review comments and reorganized the entire article. The article could be accepted.